# Antimicrobial Properties of Compounds Isolated from *Syzygium malaccense* (L.) Merr. and L.M. Perry and Medicinal Plants Used in French Polynesia

**DOI:** 10.3390/life12050733

**Published:** 2022-05-14

**Authors:** Camille Quenon, Thierry Hennebelle, Jean-François Butaud, Raimana Ho, Jennifer Samaillie, Christel Neut, Tamatoa Lehartel, Céline Rivière, Ali Siah, Natacha Bonneau, Sevser Sahpaz, Sébastien Anthérieu, Nicolas Lebegue, Phila Raharivelomanana, Vincent Roumy

**Affiliations:** 1UMRT 1158 BioEcoAgro, Métabolites spécialisés d’origine végétale, Université de Lille, JUNIA, Université de Liège, Université de Picardie Jules Verne, 59006 Lille, France; camille.quenon@icloud.com (C.Q.); thierry.hennebelle@univ-lille.fr (T.H.); jennifer.samaillie@univ-lille.fr (J.S.); celine.riviere@univ-lille.fr (C.R.); ali.siah@yncrea.fr (A.S.); natacha.bonneau@univ-lille.fr (N.B.); sevser.sahpaz@univ-lille.fr (S.S.); 2Consultant in Forestry and Polynesian Botany, BP 52832, 98716 Pirae, Tahiti, French Polynesia; jfbutaud@hotmail.com; 3EIO, UMR 241, Université de la Polynésie Française, BP 6570, 98702 Faa’a, Tahiti, French Polynesia; raimana.ho@upf.pf (R.H.); tamatoa.lehartel@upf.pf (T.L.); phila.bianchini@upf.pf (P.R.); 4Inserm U1286, Laboratoire de Bactériologie, CHU Lille, Faculté des Sciences Pharmaceutiques et Biologiques, Université Lille Nord de France, 59006 Lille, France; christel.neut@univ-lille.fr; 5UMR-S1172, Neuroscience & Cognition INSERM, Faculté des Sciences Pharmaceutiques et Biologiques, Université Lille Nord de France, CEDEX, 59006 Lille, France; sebastien.antherieu@univ-lille.fr; 6ULR 4483-IMPECS-IMPact de l’Environnement Chimique sur la Santé humaine, CHU Lille, Institut Pasteur de Lille, Université de Lille, 59006 Lille, France; nicolas.lebegue@univ-lille.fr

**Keywords:** Society Islands, traditional medicine, phytochemical, antimicrobial, salicylic compounds, *Syzygium malaccense*

## Abstract

A preliminary ethnopharmacological survey, achieved in French Polynesia, led to the collection of the most cited plants among 63 species used to treat “infectious” diseases, with a description of their medicinal uses. Bibliographical investigations and antimicrobial screening permitted the selection of the botanical species *Syzygium malaccense* (Myrtaceae) for phytochemical analysis. Leaves of *Syzygium malaccense* were usually used in mixture with rhizomes of *Curcuma longa* to treat infectious diseases such as cystitis. The methanolic plant extracts were tested in vitro with an agar microdilution method on 33 bacteria strains and 1 yeast to obtain their Minimal Inhibitory Concentration (MIC), and cytotoxicity against HepG2 cells were evaluated. Antimicrobial synergistic effects of methanolic plant extracts from leaves of *Syzygium malaccense* and rhizomes from *Curcuma longa* were also evaluated. The bio-guided isolation of leaf extract from *Syzygium malaccense* led to the identification of seven alkyl-salicylic acids (anacardic acids or ginkgolic acids C15:0, C15:1, C17:0, C17:1, C17:2, C17:3 and C19:1) described for the first time in this species. All compounds were tested against *Staphylococcus aureus* (18.75 < MIC < 75.0 µg/mL), *Streptococcus pyogenes* (2.34 < MIC < 18.75 µg/mL) and *Pseudomonas aeruginosa* (MIC = 150 µg/mL), and their structure–activity relationships were discussed. The methanolic extract and salicylic derivatives from *S. malaccense* showed an interesting antimicrobial activity against Gram+ bacteria, without toxicity on hepG2 cells at 400 μg/mL. Moreover, these antibacterial compounds have already been studied for their anti-inflammatory activity, which supports the therapeutic interest of *S. malaccense* against infectious diseases.

## 1. Introduction

In 2011, the WHO (World Health Organization) asked for an increased search for new drugs as antibiotic resistance has increased dramatically, but only a few new molecules are in development. Plants can be a source of new antimicrobial drugs, hopefully safe both for human use and the environment [1]. An ethnopharmacological study performed in the Raiatea and Tahaa islands led to the collection of 30 of the most common plants and lichens among 63 species used to treat infectious diseases. In vitro assays revealed the antimicrobial activities of 13 plants and one lichen species with MIC ≤ 0.6 mg/mL for one or several of the 34 micro-organisms. Cytotoxic assays performed on HepG2 cells did not reveal growth inhibitions at MIC ≤ 0.4 mg/mL except for *Curcuma longa* (rhizome), *Calophyllum inophyllum* (bark and leaves) and *Hibiscus tiliaceus* (bark), which were known to have specific toxicity against hepatoma cell lines (HepG2) without any effects on healthy human cells [2,3].

Then, authors focussed on *Syzygium malaccense*, which was shown to be one of the most active of the tested plants and therefore deserves deeper investigation as it was little previously reported in literature.

*S. malaccense* (Myrtaceae) is a cultivated fruit tree from French Polynesia, the leaves of which are used in local medicine to treat infectious diseases like bronchitis, cystitis, furunculosis, oral thrush, and vaginal mycosis. Scientific references also report a traditional use against a wide variety of inflammatory conditions in Western Samoa [4,5]. Nonetheless, *S. malaccense* is still poorly studied regarding the bioactive compounds of its leaves [6,7] and its antimicrobial activity [8]. 

This study reports the antimicrobial activity of methanolic extract of *S. malaccense* and of 29 other species used against infectious diseases. It also revealed the synergistic effect of the *S. malaccense* leaf extract with *Curcuma longa* rhizome extract. 

Bio-guided isolation and identification of compounds were performed by various chromatographic methods and by spectroscopic analysis (UHPLC-UV-MS, HRMS, EI-MS, NMR). Seven antibacterial compounds (6-alkenyl or 6-alkyl salicylic acids) were isolated from *S. malaccense* methanolic leaf extract and were identified for the first time in this species (Figure 1). The antimicrobial activity of some of these compounds has already been briefly mentioned in other studies against Gram-positive bacteria—*Staphylococcus aureus* [9], *Bacillus subtilis*, *Brevibacterium ammoniagenes*, *Streptococcus mutans, Cutibacterium acnes* [10] and *Porphyromonas gingivalis* [11]—and a Gram-negative bacterium (*Helicobacter pylori,* [12], but not against most of the microorganisms cited in this study, which also allowed a discussion on the structure–antibacterial activity relationships.

## 2. Results and Discussion 

### 2.1. Ethnopharmacological Survey

The survey was focused on plants and remedies used against infectious diseases in the Raiatea and Tahaa islands during 2017–2018. This preliminary step led to the identification of 63 species belonging to 36 botanical families with the collection of the 30 most cited species by 18 healers (collected species with their traditional use are presented in Table 1 and other plant species were added in Appendix A). 

*S. malaccense* leaves were usually used to treat “cystitis” (83.3% of uses) in association with *C. longa* rhizome, suggesting synergistic effects of plant extracts. 

### 2.2. Antimicrobial Activity and Cytotoxicity of Plant Methanolic Extracts 

The antimicrobial activity of 33 extracts was evaluated against a panel of 34 pathogenic and multi-resistant microbes (Table 2)**.** As the authors are aware of the difficulties in ascribing any specific identified bacterial species to a so-called “traditional use”, they decided to test plant extracts on a large panel of Gram-positive, Gram-negative bacteria and one yeast. According to previous publications concerning the anti-infective potential of natural products, plant extracts were considered to be interesting antimicrobial agents at MIC ≤ 0.30 mg/mL, as was observed eight 8 plant species: (*Calophyllum inophyllum, Cordia subcordata*, *Curcuma longa, Psidium guajava*, *Syzygium cumini*, *Syzygium malaccense, Thespesia populnea*, *Torenia crustacea*).

The screening allowed the identification of 15 plant extracts, with antibacterial activities characterised by MIC ≤ 0.6 mg/mL (Table 2 and Table 3). The higher antifungal activities against the yeast *Candida albicans* were observed for the bark extract of *Syzygium cumini* and the aerial part extract of *Torenia crustacea* (MIC = 0.15 mg/mL). 

Regarding antibacterial effects, the best activities were observed for *Calophyllum inophyllum* bark extract, mainly against Gram-positive bacteria belonging to the genus *Staphylococcus* (0.07 < MIC < 0.30 mg/mL), *Streptococcus* (MIC = 0.15 mg/mL) and the species *Corynebacterium striatum* (MIC = 0.15 mg/mL). Moderate antimicrobial activities against Gram-positive bacteria (0.15 ≤ MIC ≤ 0.30 mg/mL) were also observed for *Thespesia populnea* bark extract (against *Staphylococcus* sp and *Streptococcus agalactiae*), *Cordia subcordata* and *Syzygium cumini* bark extracts (against *Staphylococcus* sp.), and with *Psidium guajava* leaf extract (against *Staphylococcus epidermidis*).

An antibiotic control using standard compounds had been previously performed and were available for each strain (Appendix A), but it is important to remember that antibiotics are pure active compounds that do not have a vegetal origin; therefore, it cannot be expected that plant extracts could have the same efficiency as antibiotics [13]. Nevertheless, those tests were performed to obtain useful information for the choice of plant remedies against infectious diseases and to extend the scope of therapeutic applications. 

No activities against tested pathogens were detected with the following plant extracts: *Musa troglodytarum* and *M. paradiasiaca* buds, *Hibiscus tiliaceus* leaves, *Ficus tinctoria* fruits and *Ficus prolixa* leaves, *Centotheca lappacea* and *Cenchrus echinatus* aerial parts, *Fagraea berteroana* flowers and leaves, *Cordyline fruticosa* buds, *Davalia solida* rhizoma and leaves, *Commelina diffusa*, *Adenostemma viscosum* and *Rorippa sarmentosa* aerial parts.

Cytotoxic assays performed on HepG2 cells did not reveal growth inhibitions at MIC ≤ 0.4 mg/mL except for *C. longa* (rhizome, IC_50_ = 33 ± 9 μg/mL), *C. inophyllum* (bark and leaves; IC_50_ =59 ± 4 and 76 ± 5 μg/mL) and *H. tiliaceus* (bark, IC_50_ = 83 ± 10 μg/mL). While this assay was intended as a preliminary screening of nonspecific cell toxicity, its results should not discourage the use of these three plants, as their toxicity seems to be specific against tumoral cells like hepatoma cell lines (HepG2) without having any effects on healthy human cells [2,3].

### 2.3. Antimicrobial Tests of Methanolic Extract from Syzygium malaccense and Curcuma longa

In vitro antimicrobial activities were characterized by the Minimal Inhibitory Concentration (MIC) and by the Fractional Inhibitory Concentration index (FIC index, Table 3).

Results exhibited important activities of *Syzygium malaccense* leaf methanolic extract against *Candida albicans*, *Staphylococcus epidermidis*, *Streptococcus pyogenes* and *Pseudomonas aeruginosa* with MIC ≤ 0.3 mg/mL. Lower activities were also observed against *Staphylococcus aureus*, *Staphylococcus warneri*, *Staphylococcus pettenkoferi*, *Corynebacterium striatum* and *Proteus mirabilis* (0.3 ≤ MIC ≤ 0.6 mg/mL). Regarding the rhizome of *Curcuma longa*, which is traditionally used in mixture with *S. malaccense*, good antimicrobial activities were also observed against *Enterococcus faecalis*, *Staphylococcus* spp., *Streptococcus agalactiae* and *Corynebacterium striatum* (0.15 ≤ MIC ≤ 0.30 mg/mL).

The FIC determination exhibited an additive effect of the mixture of *Syzygium malaccense* (leaves) and *Curcuma longa* (rhizome), characterized by an FIC index between 0.7 and 0.9, against *Enterococcus faecalis* (C159-6), *Staphylococcus aureus* (8146, 8241, ATCC 538, T28-1 et T17-4) and *Corynebacterium striatum* (T40A3). This mixture also exhibited a synergic effect against *Pseudomonas aeruginosa* 8129, characterized by an FIC index = 0.5. 

These results suggested the presence of antimicrobial compounds in *S. malaccense* leaf methanolic extracts, which could have synergic effects with compounds from the rhizome of *Curcuma longa* such as curcumin, and reinforce the therapeutic choice of this plant mixture in traditional medicine. 

### 2.4. Structural Identification 

Activity-guided isolation of *S. malaccense* leaf extract led to the identification of seven 6-alkenyl or 6-alkyl-salicylic acids (anacardic and ginkgolic acids) that were described for the first time in this species (Figure 1). Nonetheless, these compounds have already been identified in other species such as *Ginkgo biloba* L. (Ginkgoaceae)*, Anacardium occidentale* L.*, Spondias mombin* L. (Anacardiaceae) and *Knema laurina* (Blume) Warb. (Myristicaceae).

The NMR spectra of these salicylic acid derivatives provided a common pattern of peaks characteristic of their aromatic structure (Table 4). In fact, the chemical shifts recorded for compound **1**, at δ_H_: 6.89 ppm (d, *J* = 8.4 Hz): H-3, δ_H_: 7.38 ppm (dd, *J* = 7.8, 8.4 Hz): H-4, and δ_H_: 6.79 ppm (d, *J* = 7.8 Hz): H-5, corresponding to the salicylic acid moiety, were similar to those obtained for the six other isolated compounds (Choi et al., 2004). The HSQC and HMBC spectra indicated the presence of four non-protonated carbon atoms including three carbons associated with the aromatic ring (δ_C_: 110.7, 147.5 and 163.5 ppm) and the carboxyl group of the salicylic acid at δ_C_: 175.7 ppm. Bibliographical data and NMR heteronuclear correlations permitted the complete structural elucidation of this aromatic part.

Regarding the olefinic chain unsaturations of all isolated compounds, the determination of the *cis* configuration was afforded by ^13^C-NMR spectra that exhibited specific chemical shifts at 25.5 ≤ δC ≤ 27.3 ppm for the allylic methylene carbons of a *cis* geometry, whereas a *trans* stereochemistry would be indicated by a downfield shift of approximately 10 ppm [14,15]. Moreover, IR spectra of an unsaturated chain of phenolic acids showed an important absorbance at 695–699 cm^−1^ characterizing the *cis* configuration, whereas no *trans* configuration-specific absorption was observed around 960–980 cm^−1^ [16].

Localisation of the double bound was supported by NMR data (2D-HMBC correlations, COSY or selective TOCSY), and by mass spectrometry analysis of the thiomethylated derivatives (from compounds: **2**, **4**, **6** and **7**). The position of the double bonds in the alkyl side chains was afforded by the EI-MS fragments of the *α,β*-bis-(methylthio)-derivatives produced by the reaction of the compounds with dimethyl disulfide [17]. The EI-MS spectra of the thiomethylated derivatives showed intense fragment ions [C_n_H_2n+1_S]^+^ corresponding to the cleavage of the terminal aliphatic chain residues at the original site of the unsaturation. For example, the presence of a fragment ion at *m*/*z*: 117 corresponding to the cleavage of [C_6_H_13_S]^+^ clearly revealed that the double bond on the alkyl side chain was originally located between the fifth and sixth carbons from the terminal methyl group, as observed for compounds **2**, **4** and **7**. (Figure 1).

#### 2.4.1. Compound **1**: Ginkgolic Acid (C15:0) 

Compound **1** was found to possess a molecular formula of C_22_H_36_O_3_ as derived from its HR EI-MS spectrum with *m*/*z*: 347.25840 [M − H]^−^. The mono and bidimensional CDCl_3_ 500 MHz ^1^HNMR spectra revealed a 6-alkyl-salicylic acid moiety as previously described. The bidimensional HSQC and HMBC spectra combined with the molecular formula also demonstrated the presence of a linear saturated carbon chain composed of 15 carbons linked by the C-6 of the aromatic ring, corresponding to the 6-pentadecyl salicylic acid. The comparison of spectroscopic data (UHPLC-MS and NMR) with the standard of the ginkgolic acid C15:0 indicated a total similarity of the compounds structure. This anacardic acid (15:0) was originally identified in *Amphipterygium adstringens* (Schltdl.) Schiede ex Standl. (Anacardiaceae) stem bark [11] and NMR chemical shift were also consistent with previous spectroscopic data provided by the *G. biloba* leaves phytochemical analysis [18]. 

#### 2.4.2. Compound **2**: Ginkgolic Acid (C15:1, ω5) 

The molecular formula C_22_H_34_O_3_ was established by HR EI-MS of the molecular ion [M − H]^−^ at *m*/*z* 345.24248. The ^1^H-NMR spectrum showed signals characteristic of the same salicylic moiety previously described for compound **1**, and NMR data of the aliphatic chain also exhibited the presence of 15 carbons but suggested one unsaturation. 

The COSY and selective TOCSY spectra showed correlations of protons from the terminal methyl of the aliphatic chain (δ_H_ = 0.91 ppm) with the aliphatic protons δ_H_ = 1.20 to 1.40 ppm and the proton δ_H_ = 2.03 ppm (carried by an *alpha* unsaturated carbon at δ_C_ = 27.2 ppm). The HMBC spectrum of the terminal methyl (δ_H_ = 0.91 ppm) also indicated correlations with methylenic carbons (δ_C_ = 22.7 and 32.2 ppm) and with the *alpha* unsaturated carbon at 27.2 ppm, but this correlation did not permit the determination of the position of the double bond. The localisation of this unsaturation was achieved by the fragmentation of the thiomethylated derivatives by GC-EI-MS. The presence of a specific fragment ion at *m*/*z* = 117.1, corresponding to [C_6_H_13_S]^+^ revealed that the double bond was located between the fifth and sixth carbons from the terminal methyl group, namely C10′–C11′: *ω5*. Other ions corresponding to the dehydrated, or decarboxylated, thiomethylated aromatic moiety *(m*/*z* = 305.2 and 277.1) supported this structural determination. This compound was originally isolated and identified in *Knema elegans* seed oil (Myristicaceae,) with similar spectrometric data [19].

#### 2.4.3. Compound **3**: Ginkgolic acid (C17:0) 

Compound **3** was found to possess a molecular formula of C_24_H_40_O_3_ suggested by its HR EI-MS spectrum with *m*/*z*: 375.28930 [M − H]^−^. NMR spectra exhibited a similar structure as observed for compound **1** with a 6-alkylsalicylic acid moiety substituted by a linear saturated carbon chain with two additional carbons (R: C_17_H_35_ instead of C_15_H_31_). This alkyl-salicylic acid derivative has already been isolated, and identified, from ginkgo leaves and cashew nuts with similar spectrometric data [20].

#### 2.4.4. Compound **4**: Ginkgolic Acid (C17:1, ω5)

HR EI-MS spectrum exhibited a molecular ion at *m*/*z*: 373.27316 ([M − H]^−^ calculated for C_24_H_38_O_3_) that corresponded to compound **3** with an additional unsaturation. NMR data of the aliphatic chain (linked by the C-6 of the aromatic ring) also attested to the presence of 17 carbons with one unsaturation. The COSY and HMBC spectra did not show correlations of the protons from the terminal methyl of the aliphatic chain (δ_H_ = 0.90 ppm) with the ethylenic group, suggesting a distant double bond from the terminal methyl, but did not allow its location. The comparison of TOCSY long range NMR experiments at different time mixtures (tm = 60 or 80 ms) exhibited minor correlations between the methyl terminal (δ_H_ = 0.90 ppm) and the methylenic proton (δ_H_ = 5.36 ppm) at tm = 80 ms. This comparison permitted us to assume the location of the double bond between aliphatic carbons C11′ and C14′, but additional spectrometric mass analysis was required for an accurate determination. The EI-MS spectra of the thiomethylated compound 4 exhibited a fragment ion at *m*/*z*: 117.1 corresponding to [C_6_H_13_S]^+^ and allowed the definitive location of the unsaturation between the fifth and sixth carbons from the terminal methyl group (*ω*5). Other fragment ions at m/z: 333.2 and 307.2 corresponding to the dehydrated and decarboxylated residual salicylic moiety afforded the location of this double bond at *ω*5. This compound has already been identified in plant species from various botanical families, such as Myristicaceae (*Knema elegans*), Anacardiaceae (*Anacardium occidentale*), Ginkgoaceae (*Ginkgo biloba*), Geraniaceae (*Pelargonium hybridum*) and their spectrometric data were in accordance with our results [19,20,21]. 

#### 2.4.5. Compound **5**: Ginkgolic Acid (C17:3; ω3, ω6, ω9) 

The molecular formula C_24_H_34_O_3_ was established by HR EI-MS of the molecular ion [M − H]^−^ at *m*/*z*: 369.24254. The ^1^H-NMR spectrum showed signals characteristic of the salicylic acid moiety previously described with 23 aliphatic and 6 olefinic protons. The presence of 24 carbons was demonstrated by the 13C NMR spectrum, and the heteronuclear 2D NMR (HSQC, HMBC) indicated the presence of 4 non-protonated carbons, 9 methine, 10 methylene, and 1 methyl. The salicylic acid accounted for five of the eight degrees of unsaturation from the molecular formula; the remaining three were attributed to three olefinic double bonds with their methine protons (δ_H_: 5.38 ppm, 6H). The 2D NMR correlations (HSQC and HMBC) experiments allowed assignment of the ^13^C-NMR spectrum signals, attesting to correlations of the carbons 7′, 10′ and 13′ at δ_C_: 25.5, 25.6 and 27.3 ppm for the allylic methylene carbons of the 3 *cis* unsaturations previously mentioned (Figure 2). 

The COSY and HMBC spectra showed correlations of protons from the methyl terminal of the aliphatic chain (δH = 0.99 ppm) with the protons δ_H_ = 2.0 ppm (COSY) and with δ_C_ = 20.6 ppm (*alpha* unsaturated carbon, HMBC) corresponding to the double bound in *ω*3. This localization of the unsaturation was afforded by GC-EI-MS analysis of the thiomethylated derivative, which exhibited the presence of a fragment ion at *m*/*z*: 89.2, characterizing a [C_4_H_9_S]^+^ moiety and corresponding to an unsaturation localized at *ω*3. Moreover, the selective TOCSY experiment performed on the signal at δ_H_ = 0.99 ppm exhibited correlations with the protons at δ_H_ = 2.09 and 2.82 ppm from the methylenes of the allylic groups, affording the position of double bonds in *ω*3 and *ω*6. The *ω*9 unsaturation was deduced from the chemical shift of the two methylenic protons at δ_H_ = 2.82 (carried by the allylic methylene carbon at δ_C_ = 25.5 ppm), which is specific of a methylene group wedged between two double bonds (*ω*6 and *ω*9), as it was also measured for the *ω*3-*ω*6 unsaturations [14].

Bibliographical data exhibited a similar compound previously isolated from the species *Spondias mombin* within double bonds located at *ω*3, *ω*6 and *ω*9 [22].

#### 2.4.6. Compound **6**: Ginkgolic Acid (C17:2; ω6, ω9) 

The spectral NMR and the molecular formula C_24_H_36_O_3_ deduced from the HR El-MS spectra data highlighted the presence of a 6-alkenylsalicylic acid with a C17 di-unsaturated side chain. The aromatic ring and carbonyl group accounted for five of the seven degrees of unsaturation from the molecular formula and the remaining two were attributed to the two olefinic double bonds with their methine protons (δ: 5.38 ppm, 4H).

Similar chemical shifts and correlations to those observed for compound 5 were observed, except for the aliphatic chain ending that did not exhibit any double bond in *ω*3. This compound was already identified in the aerial parts of *Spondias mombin* and their spectrometric data were corroborated [22].

#### 2.4.7. Compound **7**: Ginkgolic Acid (C19:1; ω5) 

Compound **7** was found to possess a molecular formula of C_26_H_42_O_3_ suggested by its HR El-MS spectrum with a molecular ion at *m*/*z*: 401.30478 [M − H]^−^. NMR spectra exhibited a similar structure to that observed for compound **4** with a 6-alkenylsalicylic acid moiety substituted by a linear mono-unsaturated carbon chain with two additional carbons (R: C_19_H_37_ instead of C_17_H_33_). Localisation of the olefinic unsaturation was allowed by the fragmentation of the thiomethylated derivatives by GC-EI-MS, providing a fragment ion at *m*/*z* = 117.1 corresponding to the cleavage of [C_6_H_13_S]^+^ and highlighting a double bond in *ω5*. Other ions corresponding to the dehydrated or decarboxylated thiomethylated aromatic moiety (*m*/*z* = 361.2 and 335.3) supported this structural determination as observed for compound **4**. This compound has already been identified in *Amphipterygium adstringens* bark [11].

### 2.5. Antimicrobical Activity of Compounds ***1*** to ***7***


MIC of isolated compounds and standard were determined on three strains: *S. aureus* 8241, *S. pyogenes* 19138 and *P. aeruginosa* 8129 (Table 5).

### 2.6. Synergistic Effects of Plant Extracts 

The evaluation of the antibacterial synergy of curcumin (major compound from *C. longa* rhizome) with isolated compounds from *S. malaccense* were performed by a checkerboard method, but no particular effect was observed (in contrast with the synergy observed for the crude plant extracts in mixture), suggesting the presence of other bioactive compounds. 

### 2.7. Structure–Activity Relationships

The best activities were observed against *S. pyogenes*. Bacteriological tests showed a decrease in the activity between C15:0 and C17:0 compounds. However, an improvement in the activity was observed between C15:0 and C15:1, and between C17:0, C17:1, C17:2 and C17:3, meaning an enhancement of antibacterial activity correlated with the number of unsaturations (the most unsaturated compound (**5**) was the most active against Gram-positive bacteria). Moreover, the activity of compound C17:1 was comparable to those observed for C19:1 (CMI = 18.75 μg/mL). Therefore, experimental data underlined evidence of an optimal chain length and number of unsaturation, promoting the antibacterial activity against *S. pyogenes*. Similar observations were made for other Gram-positive bacteria (*S. aureus*). Those results were in accordance with previous data regarding the importance of the number of unsaturation even if optimal activity were also previously recorded against Gram-positive or -negative bacteria for saturated alkyl chains ranged between 6 and 13 carbons [10].

Nonetheless, no important variation in the activity against *P. aeruginosa* was observed for the different compounds, suggesting another pharmacological mechanisms against this Gram-negative bacterium (ginkgolic acids could restrict *P. aeruginosa* virulence by abolishing pyocyanin pigment production) [23].

Those results attested the importance of the aliphatic chain length and its unsaturation number or location for an optimal antimicrobial activity of these salicylic derivatives against Gram-positive bacteria and permitted us to explain the antimicrobial plant extract activity.

## 3. Conclusions

This preliminary ethnopharmacological survey was the first performed in the Leeward Islands of Society archipelago (most of our previous works were focussed on the Windward Islands archipelago) and led to the selection of antimicrobial plant extracts [24,25,26]. 

*S. malaccense* leaf methanolic extract was studied for its antimicrobial activities against Gram-positive bacteria (MIC < 0.6 mg/mL) with additive or synergistic effects with *C. longa* rhizome methanolic extract. However, no antibacterial synergy was observed between ginkgolic acids and curcumin, which are, respectively, major active compounds from these two plant extracts, suggesting the presence of other bioactive compounds. The 6-alkenyl or alkyl-salicylic acids (anacardic acids or ginkgolic acids C15:0, C15:1, C17:0, C17:1, C17:2, C17:3 and C19:1) were identified as the main antibacterial compounds from the methanolic extract of *S. malaccense.* The presence of these compounds can justify the medicinal uses of the plant species against infectious diseases and creates interesting anti-inflammatory and complementary effects to those observed for similar salicylic derivatives [27] and to those described in other traditional uses and experimental data [4]. 

Antimicrobial activity of the seven isolated compounds from *S. malaccense* was determined against three strains. The present study showed a good antimicrobial activity of all compounds on *S. pyogenes*, followed by *S. aureus*, and low activity against *P. aeruginosa*. The structure–activity relationship analyses concluded to the importance of the length of the side chain of molecules as well as the number of their unsaturations. 

The methanolic extract from *S. malaccense* appeared to have interesting antimicrobial activity against Gram-positive bacteria and did not show toxicity on hepG2 cells at 400 μg/mL. Nonetheless, it is important to point out that the potential toxicity of these compounds may induce dermatitis and hepatic or renal disorders [28,29].

## 4. Materials and Methods

### 4.1. Ethnopharmacological Survey

The Tahaa and Raiatea islands are located in the Society archipelago (16°44′00″ S; 151°27′00″ W and 16°36′58″ S; 151°30′00″ W). The ethnopharmacological survey was carried out among the 18 traditional healers specialized in the treatment of “ma’i mau” or “real diseases” using semi-structured interviews. The study was focused on infectious diseases and was conducted with a rigorous ethical approach. The legal authorities of Polynesia were contacted and informed of the survey. The project was declared to the environment department of Polynesia (DIREN), with declaration to the Minister of Health and Social Services, and all informant consents were collected.

The scientific identification and biogeographical status of each plant species was checked by the botanist J.F. Butaud in accordance with regional floras [30] and traditional medicine literature [24,25,26,30,31,32]. Plant names have been checked with the “http://www.theplantlist.org” website (accessed on 3 March 2022).

### 4.2. Antimicrobial Activity 

#### 4.2.1. Plant Material and Preparation of Extract

The parts of the plants traditionally used were collected and dried in a dark room at 30 °C, then ground and macerated in methanol during 12 h (1 g/10 mL). Extracts were filtered, and the solvent was removed by evaporation under reduced pressure. 

#### 4.2.2. MIC and FIC Index Measures

The antimicrobial test of methanolic extract was carried out on Petri dishes containing Mueller-Hinton Agar (MHA) mixed with plant extract to obtain final concentrations ranged from 1.25 to 0.07 mg/mL. According to previous publications about the anti-infective potential of natural products, plant extracts were considered as active at MIC ≤ 0.30 mg/mL [33,34]. 

Extracts from *S. malaccense* and *C. longa* were tested against 34 clinical microbial strains. A multi-headed inoculator seeded the bacterial strains on prepared Petri dishes at 10^5^ CFU/mL (Colony Forming Unit/mL) in Cysteine Ringer (CR) solution (Merck, Darmstadt, Germany). Minimal Inhibitory Concentrations (MIC) were visually determined after 24 h of incubation at 37 °C. MIC values of extracts and controls were recorded as the lowest concentrations of extracts, showing no growth of colonies. The results were obtained from the average of triplicate measurements.

Synergy was evaluated by a checkerboard method and determination of the FIC index (fractional inhibitory concentration). The final extract concentrations were ranged from 1.25 to 0.07 mg/mL both for vegetal extract or compounds (A and B). FIC index was calculated with FIC = (MIC of A in combination/MIC of A alone) + (MIC of B in combination / MIC of B alone). Results were interpreted as “synergistic activity” (FIC ≤ 0.5), “additional activity” (0.5 < FIC < 1), “indifferent activity” (1 ≤ FIC ≤ 2) and “antagonistic activity” (2 < FIC) [35].

#### 4.2.3. MIC Determination of Isolated Compounds and Standard Molecules Using a Broth Microdilution Method

A serial dilution technique using 96-well microtiter plates was used to determine the MIC of the pure compounds against sensitive bacteria: *Streptococcus pyogenes* 19138, *Staphylococcus aureus* 8241 and *Pseudomonas aeruginosa* 8129. Nine concentrations of each compound, from 300 to 1.7 μg/mL, were used. They were serially twofold diluted with RC in nine wells. Two wells were represented as bacteria culture control (positive control) and medium sterility control (negative control). Then the wells were loaded with MH liquid medium and bacterial suspension (10^4^ bacteria/mL), giving a final volume of 200 μL. The plates were incubated overnight at 37 °C. Bacterial growth was indicated visually and then by direct spray of 0.2 mg/mL INT to each well with incubation at 37 °C for 30 min. Bacterial growth was indicated by a reddish-pink colour. MIC values were determined as the lowest concentrations of compounds showing clear wells. 

### 4.3. Cytotoxic Activity of Crude Plant Extracts

Hepatocellular carcinoma cell line (HepG2) and an MTT (3-(4,5-dimethylthiazol-2-yl)-2,5-diphenyltetrazolium bromide) assay method were used to quantify cell viability. Cells were seeded into a 96-well microplate in a Gibco Dulbecco’s Modified Eagles Medium (DMEM), with 10% FBS (Foetal Bovine Serum, ThermoFisher Scientific, Merelbeke, Belgium) and antibiotics (mixture penicillin/streptomycin 100 UI/mL, Sigma Aldrich, Machelen, Belgium). After two days at 37 °C (5% of CO_2_), wells were emptied by suction and the cells were treated with plant extracts diluted in the culture medium (with 0.2% DMSO, at concentrations from 400 μg/mL to 25 μg/mL), to obtain a final volume of 100 μL. After two days, the medium was discarded and replaced with DMEM containing 0.5 mg/mL MTT (3-(4,5-dimethylthiazol-2-yl)-2,5-diphenyltetrazolium bromide). After 1 h 30 min of incubation at 37 °C and 5% CO_2_, the water-insoluble formazan was dissolved in 100 μL DMSO and the absorbance was measured at 550 nm using a UV-spectrometer (Tecan, Spark 10M). The cytotoxicity of the crude extracts and standard (camptothecin: IC_50_: 0.9 ± 0.1 μg/mL) was determined by comparing the absorbance of treated cells with the absorbance of control cells cultured in 0.2% DMSO. Data were expressed as a percentage of inhibition calculated according to the formula (% Cell viability = Abs treated cells × 100/Abs control cells) and were analysed by linear regression using GraphPad Prism (5.01).

### 4.4. Bioautography and Isolation of Aliphatic-Salicylic Acids from Syzygium malaccense

#### 4.4.1. Bioautography 

Bioautography of *Syzygium malaccense* leaf methanolic extract against *Streptococcus pyogenes*, *Streptococcus pyogenes* and *Staphylococcus aureus* enabled the activity-guided isolation of antibacterial compounds. TLC was carried out on silica gel 60 F254 aluminium plates (Merck) submitted to Tol/AcOEt/MeOH (7:2:1; *v*/*v*) eluent with UV detection 254 and 366 nm. The plates were covered by MHA containing the bacterial strain suspension in Petri dishes. After incubation (24 h at 37 °C), growth was revealed by iodonitrotetrazolium chloride (INT, 2 mg/mL) and inhibition zones were visually localized on the TLC.

#### 4.4.2. Purification of Salicylic Acids Derivatives

The methanolic dried extract (50 g), obtained from 600 g of powdered leaf of *S. malaccense*, was subjected to column chromatography on silica gel (MN Kieselgel 60, 0.063–0.2 mm) and successively eluted with n-Hex, Tol, AcOEt and MeOH to afford 46 fractions. Fractions F26 to F43 were subjected to various column chromatography on silica gel (MN Kieselgel 60, 0.063–0.2 mm) and successively eluted with Tol, AcOEt and MeOH for the purification of active compounds monitored by UHPLC-UV-MS. 

Final purifications were performed by preparative-HPLC (U5SC18HQ 250 × 0.46) at a 17 mL/min flow rate; eluted with gradient H_2_O/ACN (5:95 with 0.1% formic acid) to yield compounds **1** (16 mg), **2** (3.4 mg), **3** (21 mg), **4** (94 mg), **5** (40 mg), **6** (13.2 mg) and **7** (158 mg). Compounds from *S. malaccense* and standards (ginkgolic acid C15:0, C15:1 and C17:1) were analysed by UHPLC-UV-MS using an Acquity UPLC H-Class Waters system equipped with a diode array detector (DAD) and an Acquity QDa ESI-Quadrupole Mass Spectrometer (software Empower 3, Wommelgem, Belgium). The stationary phase was a Waters Acquity BEH C18 column (2.1 × 50 mm, 1.7 μm) and the mobile phase consisted of H_2_O/ACN (10:90 with 0.1% formic acid) at 0.3 mL/min. The ionization was performed in a negative mode with a cone voltage at 10 V. 

### 4.5. Identification of Salicylic Acids Derivatives

#### 4.5.1. Spectrometric Analysis

Compounds isolated from *S. malaccense* were analysed by High Resolution Mass Spectrometry (HR-MS) using a Thermo Fisher Scientific Orbitrap Mass Spectrometer with an electrospray ion source (negative mode) and Xcalibur software (Arlington, VA, USA).

El-MS spectra of thiomethylated derivatives were realised on a GC-MS Trace DSQ (ThermoFisher in CHCl_3_, 70 eV, positive mode).

NMR spectra were recorded on a Bruker AVANCE 500 spectrometer (1H at 500 MHz- and 13C-NMR) in CDCl_3_ using standard pulse sequences.

#### 4.5.2. Thiomethylation 

To a solution of alkenyl-salicylic acid (1 eq., 0.01 mmol) in diethyl ether (2 mL) was added iodine (9.5 eq., 24 mg, 0.095 mmol) and dimethyl disulfide (450 eq., 0.4 mL, 4.5 mmol) in a sealed tube. The reaction mixture was stirred at 50 °C for 5 h, then cooled to room temperature, poured into 5% aqueous sodium thiosulfate solution (5 mL) and extracted with diethyl ether (3 mL) three times. The combined organic layer was concentrated in vacuo to give a crude dimethyldisulfide (DMDS) adduct, which was addressed to EI MS analysis without further purification. 

#### 4.5.3. Compound Spectroscopic Data

Compound **1**: 6-(pentadecyl)-salicylic acid (anacardic acid or ginkgolic acid C15:0): white wax; IR *λ*max: 2915, 2846, 1652, 1604, 1446, 1110, 1216, 706, 548 cm^−1^; UV (MeOH) *λ*max: 207.1, 243.7, 311.5 nm; ^1^H NMR (500 MHz, CDCl_3_, in ppm): δH: 7.38 (1H, dd, *J* = 8.4, 7.3 Hz), 6.89 (1H, d, *J* = 8.4 Hz), 6.80 (1H, d, *J* = 7.3 Hz), 3.00 (2H, t, *J* = 6.4 Hz), 1.60 (2H, m), 1.20 to 1.40 (24H, m), 0.90 (3H, t, *J* = 6.9); ^13^C NMR (500 MHz, CDCl_3_): δC: 175.0, 163.8, 148.0, 135.4, 122.8, 116.4, 110.5, 36.2, 31.9, 32.2, 29.0 à 29.7, 22.7 and 14.4; HR El-MS (negative) *m*/*z*: 347.25840 ([M − H]^−^ calcd for C_22_H_36_O_3_).

Compound **2**: 6-(10′Z-Pentadecenyl)-salicylic acid: ginkgolic acid C15:1 *ω5*: white wax, IR *λ*max: 2918, 2847, 1648, 1607, 1450, 1297, 1218, 720, 697 cm^−1^; UV (MeOH) *λ*max: 207.1, 243.7, 310.3 nm; ^1^H NMR (500 MHz, CDCl_3_, in ppm): δH: 7.38 (1H, dd, *J* = 7.3, 8.4 Hz), 6.89 (1H, d, *J* = 8.4 Hz), 6.80 (1H, d, *J* = 7.3 Hz), 5.38 (2H, m), 3.00 (2H, t, *J* = 6.4 Hz), 2.03 (4H, m), 1.60 (2H, m), 1.20 to 1.40 (16H, m), 0.91 (3H, t, *J* = 7.6 Hz); ^13^C NMR (500 MHz, CDCl_3_): 175.7, 163.8, 148.0, 135.4, 130.3, 122.8, 116.4, 110.5, 36.2, 32.2, 31.9, 29.0 to 29.7, 27.3, 27.2, 22.7 and 14.4 ppm; HR El-MS: m/z: 345,24254 [M − H]^−^ calcd for C_22_H_34_O_3_.

Compound **3**: 6-(Heptadecyl)-salicylic acid: ginkgolic acid C17:0: white wax; IR *λ*max: 2910, 2842, 1653, 1601, 1448, 1109, 1212, 704, 547 cm^−1^; UV (MeOH) *λ*max: 215.3, 243.7, 310.3; ^1^H NMR (500 MHz, CDCl_3_, in ppm): δH: 7.38 (1H, dd, *J* = 7.5, 8.2 Hz), 6.89 (1H, d, *J* = 8.2 Hz), 6.80 (1H, d, *J* = 7.5 Hz), 3.00 (2H, 6.7), 1.60 (2H, m), 1.20 to 1.40 (28H, m), 0.90 (3H, t, *J* = 6.9 Hz); ^13^C NMR (500 MHz, CDCl_3_): δC: 175.7, 163.5, 147.5, 135.4, 122.8, 116.4, 110.5, 36.2, 32.2, 31.9, 29.7 to 29.0, 22.7 and 14.1 ppm; HR El-MS: *m*/*z*: 375.28930 ([M − H]^−^ calcd for C_24_H_40_O_3_).

Compound **4**: 6-(10′Z-Heptadecenyl)-salicylic acid: ginkgolic acid C17:1 *ω5*: white wax; IR *λ*max: 2916, 2848, 1649, 1606, 1445, 1297, 1218, 720, 698 cm^−1^; UV (MeOH) *λ*max: 215.3, 243.7, 310.3 nm; ^1^H NMR (500 MHz, CDCl_3_, in ppm): δH: 7.39 (1H, dd, *J* = 7.8, 8.4 Hz), 6.89 (1H, d, *J* = 8.4 Hz), 6.80 (1H, d, *J* = 7.8 Hz), 5.36 (2H, m), 3.00 (2H, t, *J* = 6.4 Hz), 2.09 (4H, m), 1.60 (2H, m), 1.20 to 1.40 (20H, m), 0.90 (3H, t, *J* = 6.9 Hz). ^13^C NMR (500 MHz, CDCl_3_): δC: 175.7, 163.5, 147.5, 135.4, 129.9, 129.9, 122.8, 115.7, 110.7, 36.5, 32.3, 31.9, 29.8 to 29.0, 27.2, 26.9, 22.6 and 14.0 ppm; HR El-MS: *m*/*z*: 373.27316 ([M − H]^−^ calcd for C_24_H_38_O_3_).

Compound **5**: 6-(8′Z,11′Z,14′Z-Heptadecatrienyl)-salicylic acid: ginkgolic acid (C17:3 *ω*3; *ω*6; *ω*9 or penladjuaic acid: colorless oil; IR *λ*max: 2932, 2856, 1647, 1608, 1448, 1297, 1218, 720, 698 cm^−1^, UV (MeOH) *λ*max 215.3, 243.7, 310.3 nm; ^1^H NMR (500 MHz, CDCl_3_, in ppm): δH: 7.38 (1H, dd, *J* = 7.8, 8.4 Hz), 6.88 (1H, d, *J* = 8.4 Hz), 6.79 (1H, d, *J* = 7.8 Hz), 5.38 (6H, m), 3.00 (2H, t, *J* = 6.4 Hz), 2.82 (4H, m), 2.09 (4H, m), 1.60 (2H, m), 1.20 à 1.40 (8H, m) and 0.99 (3H, t, *J* = 7.6 Hz); ^13^C NMR (500 MHz, CDCl_3_): δC: 175.7, 163.5, 147.5, 135.0, 132.5, 132.0, 130.4, 128.3, 127.6, 127.2, 122.8, 115.7, 110.7, 36.5, 32.3, 29.7, 29.6, 29.4, 29.3, 27.3, 25.5, 25.6, 20.6 and 14.7 ppm; HR El-MS: *m*/*z*: 369.24254 ([M − H]^−^ calcd for C_24_H_34_O_3_).

Compound **6**: 6-(8′Z,11′Z-Heptadedienyl)- salicylic acid:ginkgolic acid or penladjuaic acid (*C17:2 ω6; ω9*): white wax; IR *λ*max: 2920, 2860, 1650, 1609, 1450, 1299, 1216, 718, 695 cm^−1^; UV (MeOH) *λ*max: 198.4, 243.7, 311.5 nm; ^1^H NMR (500 MHz, CDCl_3_, in ppm): δH: 7.38 (1H, dd, *J* = 7.8, 8.2 Hz), 6.88 (1H, d, *J* = 8.2 Hz), 6.79 (1H, d, *J* = 7.8 Hz), 5.38 (4H, m), 3.00 (2H, t, *J* = 7.8 Hz), 2.82 (2H, t, *J* = 6.7 Hz), 2.07 (4H, m), 1.60 (2H, m), 1.31, 1.20 to 1.40 (14H, m), 0.89 (3H, t, *J* = 6.7 Hz); ^13^C NMR (500 MHz, CDCl_3_): δC: 175.7, 163.5, 147.5, 135.0, 130.2, 130.1, 128.0, 127.9, 122.8, 115.7, 110.7, 36.5, 32.3, 31.5, 29.7, 29.6, 29.4, 29.4, 29.3, 27.2 (2C), 25.5, 22.6 and 14.3 ppm; HR El-MS: *m*/*z*: 371.25884 ([M − H]^−^ calcd for C_24_H_36_O_3_).

Compound **7**: 2-(14-Nonadecenyl)-6-hydroxybenzoic acid: ginkgolic acid C19:1 *ω*5: white wax IR *λ*max: 2916, 2848, 1652,1605, 1446, 1297, 1168, 721, 698 cm^−1^; UV (MeOH) *λ*max: 208.2, 243.7, 310.3 nm; ^1^H NMR (500 MHz, CDCl_3_ in ppm): δH: 7.39 (1H, dd, *J* = 7.7, 8.0 Hz), 6.90 (1H, d, *J* = 7.7 Hz), 6.81 (1H, d, *J* = 8.0 Hz), 5.38 (2H, m), 3.00 (2H, m), 2.04 (4H, m), 1.63 (2H, m), 1.20 to 1.40 (24H, m), 0.93 (3H, t, *J* = 6.8 Hz); ^13^C NMR (500 MHz, CDCl_3_): δC: 176.4, 163.5, 147.8, 135.4, 129.8, 129.8 122.7, 115.8, 110.5, 36.5, 32.0, 31.9, 29.8 to 29.3, 27.9, 27.7, 22.3 and 14,0 ppm; HR El-MS: *m*/*z*: 401.30478 ([M − H]^−^ calcd for C_26_H_42_O_3_).

## Figures and Tables

**Figure 1 life-12-00733-f001:**
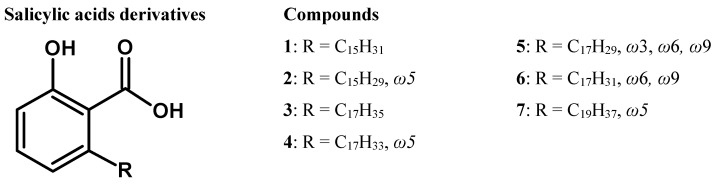
Aliphatic-salicylic acid derivatives isolated from the leaves of *Syzygium malaccense.*

**Figure 2 life-12-00733-f002:**
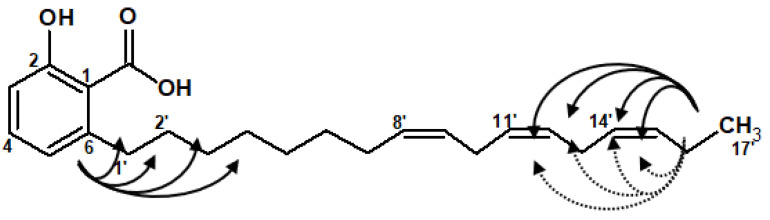
Heteronuclear 2D NMR correlations (HMBC in bold) and homonuclear selective TOCSY correlations (dotted line) of the aliphatic chain from compound **5**.

**Table 1 life-12-00733-t001:** Medicinal plants most cited in the survey with their medicinal uses.

Latin Name “Vernacular Name” (Botanical Family) (Number of Citation)	Health Problem Category (Part Used; Mode of Administration)
*Adenostemma viscosum* J.R.Forst. & G. Forst. “vaianu” (Asteraceae) (5)	**Pneumo.** (l.; Po), **Uterus** (l.; Po), **Cystitis** (l.; Po), **V. Cutaneous symptoms** (w.p.; Po), **V. diseases** (l.; Po and Cat), **Wound, Abscess** (l.; Po)
*Aleurites moluccanus* (L.) Willd. “ti’a’iri” (Euphorbiaceae) (9)	**Aphta** (bk.; Po), **Cystitis** (bk.; Po), **Furunculous, Pustule** (alm.; Bath), **STD** (bk.; Po), **Oral mycosis** (bk.; Garg and Po), **Tonsillitis** (bk. Garg or Po and Cat.), **Wound, Abscess** (bk.; Po)
*Calophyllum inophyllum* L. “tamanu” (Calophyllaceae) (8)	**Cicatrising** (alm. (oil); Cat), **Cystitis** (bk.; Po), **Mycosis** (l.; Bath), **V. Cutaneous symptoms** (l.; Bath), **Wound, Abscess** (alm. (oil); Cat) (bk.; Po)
*Cenchrus echinatus* L. “piripiri” (Poaceae) (7)	**Furunculous, Pustule** (a.p.; Po), **Leucorrhoea** (a.p.; Po), **Sinusitis, Rhinitis** (a.p.; Fumigation), **Wound, Abscess** (a.p.; Cat)
*Centotheca lappacea* (L.) Desv. “’ofe’ofe” (Poaceae) (7)	**Furunculous, Pustule** (a.p.; Po), **V. diseases** (a.p.; Po and Cat), **Wound, Abscess** (a.p.; Cat)
*Citrus**× aurantiifolia* (Christm.) Swingle “taporo” (Rutaceae) (13)	**Pneumo.** (f. juice or l.; Po), **Uterus**, (f. juice; Po), **Cystitis** (f. juice or l.; Po), **Diarrhoea** (f. juice; Po or Po and Cat.), **STD** (f. juice; Po), **Heartburn** (f. juice; Po), **Leucorrhoea** (f. juice or l.; Po), **Oral mycosis** (f. juice; Po), **Tonsillitis** (f. juice; Po), **V. Cutaneous symptoms** (f. juice; Cat), **Vulvitis** (f. juice; Po), **Wound, Abscess** (f. juice; Cat)
*Commelina diffusa* Burm.f. “ma’a pape” (Commelinaceae) (6)	**Wound, Abscess** (l. and st.; Cat)
*Cordia subcordata* Lam. “tou” (Boraginaceae) (12)	**Acne** (l.; ND), **Pneumo.** (bk. or l.; Po), **Conjunctivitis** (l.; In eye), **Cystitis** (bk. or l.; Po), **Furunculous, Pustule** (bk.; Po), **STD** (l.; Po), **Leucorrhoea** (bk.; Po), **Oral mycosis** (bk. orl.; Po), **Sinusitis, Rhinitis** (l.; Po), **V. Cutaneous symptoms** (l.; Cat), **Wound, Abscess** (bk.; Po)
*Cordyline fruticosa* (L.) A.Chev. “‘auti” (Asparagaceae) (5)	**Diarrhoea** (b.; Po) (l.; Po and Cat), **Heartburn** (l.; Po), **Otitis** (b.; Cat), **V. Cutaneous symptoms** (b.; Cat)
*Curcuma longa* L. “re’a ma’a” (Zingiberaceae) (15)	**Pneumo.** (rh.; Po), **Uterus** (rh.; Po), **Cystitis** (rh.; Po, genital bath), **Furunculous, Pustule** (rh.; Po), **STD** (rh.; Po genital bath) **Oral mycosis** (rh.; Po), **Vaginal Mycosis** (rh.; Po and Bath), **V. Cutaneous symptoms** (rh.; Po), **Vulvitis** (rh.; Po)
*Davallia solida* (G. Forst.) Sw. var. *solida* “titi” (Davalliaceae) (9)	**“Crumpled lung”** (rh.; Po), **Pneumo.** (Frond or rh.; Po), **Cystitis** (rh.; Po), **Heartburn** (rh.; Po), **Leucorrhoea** (rh.;Po), **Tonsillitis** (rh.; Po), **V. Cutaneous symptoms** (rh.; Cat)
*Fagraea berteroana* A.Gray ex Benth. “pua” (Gentianaceae) (6)	**Itching, Allergy** (fl.; Bath)
*Ficus prolixa* G.Forst. “ ‘ora” (Moraceae) Ind. (6)	**Pneumo.** (a.r.; Po), **STD** (a.r.; Po), **Ringworm** (l.; Bath), **Tonsillitis** (a.r. Garg), **V. disease** (a.r.; Po and Cat)
*Ficus tinctoria* G.Forst. subsp. *tinctoria* “mati” (Moraceae) (5)	**Acne** (l.f.; ND), **Uterus** (l.f.; Po), **Cystitis** (l.f.; Po), **Furunculous, Pustule** (l.f.; Po), **Oral mycosis** (l.f.; Po)
*Hibiscus tiliaceus* L. subsp. tiliaceus “purau” (Malvaceae) (5)	**Eye irritation** (r.; In eye), **V. Cutaneous symptoms** (bk. Bath), **Wound; Abscess** (fl.; Bath)
*Mangifera indica* L. “vi popa’a” (Anacardiaceae) (5)	**Pneumo.** (f.; Po), **Furunculous, Pustule** (f.; Po and Cat), **Tonsillitis** (f.; Garg and Po), **Wound, Abscess** (f.; Po and Cat)
*Microsorum grossum* (Langsd. & Fisch.) S.B.Andrews “metuapua’a” (Polypodiaceae) (10)	**“Crumpled lung”** (rh.; Po), **Pneumo.** (rh.; Po), **Cystitis** (rh.; Po), **Diarrhoea** (Frond: Po), **Furunculous, Pustule** (rh.; Po), **Heartburn** (rh.; Po), **Leucorrhoea** (rh.; Po), **Tonsillitis** (rh.; Po), **V. Cutaneous symptoms** (rh.; Cat)
*Musa troglodytarum* L. “fe’i” (Musaceae) (5)	**V. Cutaneous symptoms** (f.;Cat), **Wound; Abscess** (b.; Cat)
*Musa**× paradisiaca* L. “mei’a hamoa” or “mei’a rio” (Musaceae) (5)	**Wound, Abscess** (b. or f.; Cat)
*Pandanus tectorius* Parkinson var. *tectorius* “fara” (Pandanaceae) (5)	**Oral mycosis** (a.r.; Po)
*Paspalum vaginatum* Sw. “matie tatahi” (Poaceae) Ind. (8)	**Aqueous pustule** (a.p.; Cat), **Pneumo.** (a.p.; Po), **Furunculous, Pustule** (a.p.; Po), **V. Cutaneous symptoms** (a.p.; Bath), **V. Diseases** (a.p.; Po and Cat)
*Phyllanthus amarus S*chumach. & Thonn. “moemoe” (Phyllanthaceae) (10)	**Furunculous, Pustule** (w.p.; Bath), **Otitis** (l.; In ear) (w.p. Cat), **V. Cutaneous symptoms** (r.; Cat)
*Psidium guajava* L. “tuvava” (Myrtaceae) (6)	**Pneumo.** (l.; Po), **Cystitis** (l.; Po) STD (l.; Po), **Otitis** (f.; In ear)
*Rorippa sarmentosa* (Sol. ex G.Forst. ex DC.) J.F.Macbr. “moahau’a’ino” (Brassicaceae) (11)	**Pneumo.** (w.p.; Po), **Uterus** (l.; Po), **Cicatrasing** (w.p.; Cat), **Constipation** (w.p.; Po), **Cystitis** (r. or w.p.; Po), **Furunculous, Pustule** (w.p.; Cat), **Heartburn** (l.; Po), **Oral mycosis** (l. Garg and Po) (r.; Po), **Purge** (w.p.; Po), **Tonsillitis** (l. or w.p.; Garg and Po or Po) (r.; Garg)
*Syzygium cumini* (L.) Skeels “pistas” (Myrtaceae) (5)	**Mycosis** (bk.; Cat), **V. Cutaneous symptoms** (bk.; Cat)
*Syzygium malaccense* (L.) Merr. & L.M.Perry “ ‘ahi’a” (Myrtaceae) (12)	**Pneumo.** (l.; Po), **Uterus** (l.; Po), **Cystitis** (bk. or l.; Po), **Furunculous, Pustule** (bk. or l.; Po), **STD** (l.; Po), **Oral mycosis** (l.; Garg or Po), **Tonsillitis** (l.; Garg and Po), **Vaginal Mycosis** (l.; Po and Bath), **Vulvitis** (l.; Po)
*Thespesia populnea* (L.) Sol. ex Corrêa “miro” (Malvaceae) (9)	**Pneumo.** (Latex of f.; Po), **Conjunctivitis** (alm.; Po), **Cystitis** (bk.; Po), **Furunculous, Pustule** (alm.; Po) (Latex of f.; Bath), **Leucorrhoea** (bk. or l.; Po), **Mycosis** (bk.; Bath), **Varicella** (Latex of f.; Bath), **Wound, Abscess** (bk.; Po)
*Torenia crustacea* (L.) Cham. & Schltdl “piri’ate” (Linderniaceae) (6)	**Pneumo.** (a.p.; Po), **Uterus** (a.p. or l. Po), **Constipation** (a.p.; Po), **Cystitis** (l.; Po), **Heartburn** (l.; Po), **Sinusitis, Rhinitis** (l.; Po), **V. diseases** (a.p.; Po and Cat)
*Usnea* spp. “remu ha’ari” (Parmeliaceae, lichen) (5)	**“Crumpled lung”** (w.p.; Po), **Pneumo.** (w.p.; Po), **Constipation** (w.p.; Po), **Cystitis** (Lichen; Po)
*Vigna marina* (Burm.) Merr. “pipi” (Fabaceae) (5)	**Burn** (l.; Bath), **Diarrhoea** (l.; Po)

Parts used: leaf (l.); fruit (f.); flower (fl.); bud (b.); bark (bk.); whole plant (w.p.); rhizome (rh.); root (r.); aerial part (a.p.); aerial root (a.r.); stem (st.); almond (alm.); latex of fig (l.f.). Route of administration-preparation: Per os (Po); Cataplasm (Cat); Gargle (Garg); Inhalation (Inh). Medicinal uses (abbreviations and specifications): **V. Cutaneous symptoms** (various symptoms: spots, rash, erysipelas, folliculitis), **Pneumo**. (bronchitis; cough; pneumonia); **V. diseases** (general infectious diseases), **STD** (Sexually transmitted disease, **Uterus** (care post-partum, “impure” menstruation), **Burn** (sunburn sometimes with headache and dehydration), **Purge** (diet between ra’au tahiti and conventional medicinal drug), “**Crumpled lung**” (crumpled lung provoking cough and pain), **Umbilic.** (care of umbilical cord).

**Table 2 life-12-00733-t002:** MIC (mg/mL) of methanolic plant extracts.

				Malvaceae	Myrtaceae	Calophyllaceae	Boraginaceae	Anacardiaceae	Parmeliaceae	Euphorbiaceae	Pandanaceae	Polypodiaceae	*Linderniaceae*
				*Hibiscus tiliaceus*	*Thespesia populnea*	*Syzygium cumini*	*Psidium guajava*	*Calophyllum inophyllum*	*Cordia subcordata*	*Mangifera indica*	*Usnea* spp.	*Aleurites moluccanus*	*Pandanus tectorius* var*. tectorius*	*Microsorum grossum*	*Torenia crustacea*
Microorganisms Tested	Bark	Bark	Leaf	Bark	Leaf	Bark	Leaf	Bark	Leaf	Lichen	Bark	Aerial Roots	Rhizome	Aerial Part
**Yeast**	*Candida albicans*	10286	1.2	0.6	1.2	0.15	0.6	0.6	1.2	0.6	1.2	1.2	-	1.2	-	0.15
**Gram +**	*Enterococcus faecalis*	C159-6	-	0.3	-	-	0.6	0.6	-	1.2	-	0.6	-	-	-	-
	*Enterococcus* sp.	8153	-	0.3	-	-	-	1.2	-	-	-	0.6	-	-	-	-
	*Staphylococcus aureus*	8146	-	0.3	1.2	0.6	0.6	0.15	0.6	0.6	1.2	0.6	-	-	-	-
	*Staphylococcus aureus*	8241	-	0.3	1.2	0.6	1.2	0.07	0.6	0.6	1.2	0.6	-	-	1.2	-
	*Staphylococcus aureus*	ATCC 6538	0.6	0.15	0.6	0.3	0.6	0.07	0.6	0.6	1.2	0.6	-	1.2	1.2	-
	*Staphylococcus aureus*	T28-1	-	0.3	0.6	0.3	0.3	0.07	0.3	0.6	1.2	0.6	-	-	1.2	-
	*Staphylococcus aureus*	T17-4	-	0.3	1.2	0.3	0.6	0.07	0.6	0.6	1.2	0.6	-	1.2	1.2	-
	*Staphylococcus epidermidis*	T46A1	0.6	0.15	0.6	0.3	0.3	0.15	0.6	0.3	1.2	0.6	0.6	-	1.2	0.6
	*Staphylococcus epidermidis*	T19A1	0.6	0.15	0.6	0.3	0.3	0.15	0.6	0.15	0.6	0.6	0.6	-	1.2	0.6
	*Staphylococcus epidermidis*	T21A5	1.2	0.3	0.6	0.3	0.3	0.15	0.6	0.15	0.6	0.6	1.2	-	1.2	0.6
	*Staphylococcus warneri*	T12A12	1.2	0.6	-	0.3	0.6	0.3	-	0.3	1.2	1.2	-	-	1.2	-
	*Staphylococcus warneri*	T26A1	1.2	1.2	-	0.3	0.6	0.3	-	0.3	0.6	-	1.2	-	1.2	-
	*Staphylococcus pettenkoferi*	T47.A6	1.2	0.3	-	0.3	0.6	0.6	-	0.6	0.6	1.2	1.2	-	1.2	-
	*Streptococcus agalactiae*	T38.2	-	0.15	1.2	-	0.6	0.15	0.6	-	1.2	0.6	-	1.2	-	0.3
	*Streptococcus agalactiae*	T53C9	-	0.15	1.2	-	0.6	0.15	0.6	0.6	-	0.6	-	1.2	-	0.6
	*Corynebacterium striatum*	T40A3	1.2	0.6	0.6	0.15	0.3	0.15	0.3	0.6	1.2	0.6	1.2	-	0.6	-
**Gram −**	*Citrobacter freundii*	11041	-	-	-	-	-	-	-	1.2	-	-	-	-	-	-
	*Citrobacter freundii*	10268	1.2	1.2	-	0.6	-	-	-	1.2	-	-	-	-	-	-
	*Proteus mirabilis*	11060	-	-	-	0.6	-	-	-	1.2	-	-	-	-	-	-
	*Proteus mirabilis*	T28-3	-	-	-	0.6	-	-	-	1.2	-	-	-	-	-	-
	*Pseudomonas aeruginosa*	8131	-	-	-	0.6	0.6	-	-	1.2	-	1.2	1.2	-	-	-
	*Pseudomonas aeruginosa*	ATCC 27583	0.6	1.2	-	0.15	0.6	-	-	0.3	1.2	0.6	0.6	-	-	0.6
	*Pseudomonas aeruginosa*	8129	0.6	1.2	1.2	0.3	0.6	1.2	1.2	0.3	1.2	1.2	1.2	1.2	-	0.6

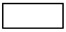
MIC: 0.6 mg/mL, 
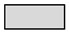
MIC: 0.6 mg/mL ≥ MIC > 0.3 mg/mL, 
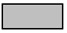
MIC ≤ 0.3 mg/mL; (-) no MIC or FIC determined at 1.2 mg/mL; (*n* = 3; with standard deviation for value: 1.2 ± 0.4 mg/mL; 0.6 ± 0.2 mg/mL; 0.3 ± 0.1 mg/mL; 0.15 ± 0.05 mg/mL; 0.07 ± 0.03 mg/mL).

**Table 3 life-12-00733-t003:** MIC (mg/mL) of methanolic extract from leaves of *S. malaccense* and rhizome of *C. longa.*

Tested Microorganisms	*Syzygium malaccense*, MIC (mg/mL)	*Curcuma longa,* MIC (mg/mL)	FIC Index	Effect
**Yeast**	*Candida albicans*	10286	0.15	0.15	1.3	Indifferent
**Gram +**	*Enterococcus faecalis*	C159-6	1.2	0.15	0.9	**Additive**
	*Staphylococcus aureus*	8146	1.2	0.3	0.8	**Additive**
	*Staphylococcus aureus*	8241	0.6	0.3	0.8	**Additive**
	*Staphylococcus aureus*	ATCC 6538	-	0.3	0.7	**Additive**
	*Staphylococcus aureus*	T28-1	0.6	0.3	0.7	**Additive**
	*Staphylococcus aureus*	T17-4	0.6	0.3	0.9	**Additive**
	*Staphylococcus epidermidis*	T46A1	0.15	0.15	1.1	Indifferent
	*Staphylococcus epidermidis*	T19A1	0.15	0.07	1.3	Indifferent
	*Staphylococcus epidermidis*	T21A5	0.15	0.07	1.2	Indifferent
	*Staphylococcus warneri*	T12A12	0.6	0.15	1.9	Indifferent
	*Staphylococcus warneri*	T26A1	0.6	0.6	1.5	Indifferent
	*Staphylococcus pettenkoferi*	T47.A6	0.6	0.6	1.5	Indifferent
	*Streptococcus agalactiae*	T53C9	1.2	0.15	1.3	Indifferent
	*Corynebacterium striatum*	T40A3	0.6	0.15	0.9	**Additive**
*Streptococcus pyogenes*	16138	0.3	-	-	-
*Streptococcus pyogenes*	19140	0.15	-	-	-
**Gram −**	*Proteus mirabilis*	11060	1.2	-	-	-
	*Proteus mirabilis*	T28-3	0.6	-	-	-
	*Pseudomonas aeruginosa*	8131	1.2	-	-	-
	*Pseudomonas aeruginosa*	ATCC 27583	0.15	-	-	-
	*Pseudomonas aeruginosa*	8129	0.15	0.6	0.5	**Synergic**

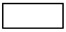
MIC: 0.6 mg/mL, 
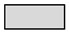
MIC: 0.6 mg/mL ≥ MIC > 0.3: mg/mL, 
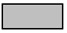
MIC ≤ 0.3 mg/mL; (-) no MIC or FIC determined at 1.2 mg/mL; (*n* = 3; with standard deviation for value: 1.2 ± 0.4 mg/mL; 0.6 ± 0.2 mg/mL; 0.3 ± 0.1 mg/mL; 0.15 ± 0.05 mg/mL; 0.07 ± 0.03 mg/mL) with FIC index of mixture.

**Table 4 life-12-00733-t004:** ^1^H and ^13^C NMR chemical shifts in ppm of compounds **1**–**7** (500 MHz in CDCl_3_).

	Comp	1	2	3	4	5	6	7
C (H)								
**1**		110.5	110.7	110.5	110.7	110.7	110.7	110.5
**2**		163.8	163.8	163.8	163.5	163.5	163.5	163.5
**3**		116.4 (6.89, d:8.4 Hz)	116.4 (6.89, d:8.4 Hz)	116.4 (6.89, d:8.2 Hz)	115.7 (6.89, d:8.4 Hz)	115.7 (6.88, d:8.4 Hz)	115.7 (6.88, d:8.2 Hz)	115.8 (6.90, d:7.7 Hz)
**4**		135.4 (7.38, d:7.3 Hz)	135.4 (7.38, dd:7.3, 8.4 Hz)	135.4 (7.38, dd:7.5, 8.2 Hz)	135.4 (7.38, dd:7.8, 8.4 Hz)	135.0 (7.38, dd:7.8, 8.4 Hz)	135.0 (7.38, dd:7.8, 8.2 Hz)	135.4 (7.39, dd:7.7, 8.0 Hz)
**5**		122.8 (6.80, d:7.3 Hz)	122.8 (6.80, d:7.3 Hz)	122.8 (6.80, d:7.5 Hz)	122.8 (6.80, d:7.8 Hz)	122.8 (6.79, d:7.8 Hz)	122.8 (6.79, d:7.8 Hz)	122.7 (6.81, d:8.0 Hz)
**6**		148.0	148.0	148.0	147.5	147.5	147.5	147.8
**1′**		36.2 (3.00, t:6.9 Hz)	36.2 (3.00, t:6.4 Hz)	36.2 (3.00, t:6.7 Hz)	36.5 (3.00, t:6.4 Hz)	36.5 (3.00, t:6.4 Hz)	36.5 (3.00, t:7.8 Hz)	36.5 (3.0, m)
**2′**		31.9 (1.60, m)	31.9 (1.60, m)	31.9 (1.60, m)	31.9 (1.60, m)	32.3 (1.60, m)	31.5 (1.60, m)	31.9 (1.63, m)
**3′–6′**		29.0–29.7 (1.20–1.40, m)	29.0–29.7 (1.20–1.40, m)	29.0–29.7 (1.20–1.40, m)	29.0–29.8 (1.20–1.40, m)	29.3–29.7 (1.20–1.40, m)	29.0–29.7 (1.20–1.40, m)	29.3–29.8 (1.20–1.40, m)
**7′**		“	“	“	“	27.3 (2.09, m)	27.2 (2.07, m)	“
**8′**		“	“	“	“	127.2–132.5 (5.38, m)	127.9–130.2 (5.38, m)	“
**9′**		“	27.3 (2.03, m)	“	“	127.2–132.5 (5.38, m)	127.9–130.2 (5.38, m)	“
**10′**		“	130.3 (5.38, m)	“	“	25.5 (2.82, m)	25.5 (2.82, t:6.7)	“
**11′**		“	130.3 (5.38, m)	“	27.2 (2.09, m)	127.2–132.5 (5.38, m)	127.9–130.2 (5.38, m)	“
**12′**		“	27.2 (2.03, m)	“	129.9 (5.36, m)	127.2–132.5 (5.38, m)	127.9–130.2 (5.38, m)	“
**13′**		32.2 (1.20–1.40, m)	32.2 (1.20–1.40, m)	“	129.9 (5.36, m)	25.6 (2.82, m)	27.2 (2.07, m)	27.9 (2.04, m)
**14′**		22.7 (1.20–1.40, m)	22.7 (1.20–1.40, m)	“	26.9 (1.20–1.40, m)	127.2–132.5 (5.38, m)	29.0-29.7 (1.20–1.40, m)	129.8 (5.38, m)
**15′**		14.4 (0.90, t:6.9 Hz)	14.4 (0.91, t:7.6 Hz)	32.2 (1.20–1.40, m)	32.3 (1.20–1.40, m)	127.2-132.5 (5.38, m)	32.3 (1.20–1.40, m)	129.8 (5.38, m)
**16′**		-	-	22.7 (1.20–1.40, m)	22.6 (1.20–1.40, m)	20.6 (2.09, m)	22.6 (1.20–1.40, m)	27.7 (2.04, m)
**17′**		-	-	14.1 (0.90, t:6.9 Hz)	14.0 (0.90, t:6.9 Hz)	14.7 (0.99, t:7.6)	14.3 (0.89, t:6.7 Hz)	32.0 (1.20–1.40, m)
**18′**		-	-	-	-	-	-	22.3 (1.20–1.40, m)
**19′**		-	-	-	-	-	-	14.0 (0.93, t:6.8 Hz)
**COOH**		175.0	175.7	175.0	175.7	175.7	175.7	176.4

**Table 5 life-12-00733-t005:** MIC (µg/mL) of compounds isolated from *S. malaccense* and standard molecules (*n* = 3); (NA: Not Active).

		MIC (µg/mL)		
Compounds	Aliphatic Chain	*Staphylococcus aureus*	*Streptococcus pyogenes*	*Pseudomonas aeruginosa*
**1**	C15:0	75.0	9.37	150
**2**	C15:1, ω5	18.75	9.37	150
**3**	C17:0	37.5	18.75	150
**4**	C17:1, ω5	18.75	9.37	150
**5**	C17:3, ω3; ω6; ω9	18.75	2.34	150
**6**	C17:2, ω6; ω9	18.75	4.68	150
**7**	C19:1, ω5	18.75	9.37	150
**Gentamycin**		0.5	0.125	0.03
**Vancomycin**		1.0	0.25	NA
**Amoxicillin**		4.0	0.03	NA

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
