# Peer review of "Antimicrobial Properties of Compounds Isolated from Syzygium malaccense (L.) Merr. and L.M. Perry and Medicinal Plants Used in French Polynesia"

_life, 2022, doi:10.3390/life12050733_

Round 1

Reviewer 1 Report

The manuscript is well written and can be published with minor corrections. Please see specific comments below:

Abstract

Results of antimicrobial test is lacking in abstract.

Please mention the clear conclusion of your study in the abstract.

Introduction

I suggest authors to add the importance of medicinal plants in maintaining human health and it is thus important to investigate them. This can be rephrased and add in the start of the introduction.

Results

Well written.

Materials and Methods

Line 429: ‘Plant species were collected, and dried ..’ Did you use whole plant material? Please mention.

Author Response

Dear Reviewer

You will find the revised manuscript, I hope it will meet your requirements :

Results of antimicrobial test with a clear conclusion were added in abstract.

The importance of medicinal plants for human health and infectious diseases were explained in the introduction.

Line 429: ‘Plant species were collected, and dried ..’ were replaced by :“The parts of the plants traditionally used were collected”

"Please see the attachment.", corrections were underlined in yellow

Reviewer 2 Report

This manuscript reported the isolation and structural elucidation of seven known alkyl-salicylic acids from some medicinal plants used in French Polynesia. All the structures were elucidated based on various spectroscopic data, including 1D, 2D-NMR, MS, etc.,  and data was compared to those reported from the literature. As claimed, these compounds were identified from the specific plant for the first time. In addition, all of them displayed  weak antimicrobial activity observed by the MIC. As far as I am concerned, this manuscript can be assigned as minor revision.

  Here are my comments.   1. Only C-NMR data was listed in the Table 4, it is highly recommended to list the H-NMR data in that table as well, even it has been reported   2. Table 5 displayed the anitmicrobial activity of seven compounds, Please include the positive control, so that the data is comparable to some known compounds.

Author Response

Dear Reviewer

You will find the revised manuscript, I hope it will meet your requirements :

Table 4 an 5 were modified, providing H NMR data and antimicrobial standards
